# Response of Global Air Pollutant Emissions to Climate Change and Its Potential Effects on Human Life Expectancy Loss

**Qianwen Cheng, Manchun Li \*, Feixue Li and Haoqing Tang**

School of Geography and Ocean Science, Nanjing University, Nanjing 210023, China
* Correspondence: limanchun@nju.edu.cn

**Abstract:** Geographical environment and climate change are basic factors for spatial fluctuations in the global distribution of air pollutants. Against the background of global climate change, further investigation is needed on how meteorological characteristics and complex geographical environment variations can drive spatial air pollution variations. This study analyzed the response of air pollutant emissions to climate change and the potential effects of air pollutant emissions on human health by integrating the air pollutant emission simulation model (GAINS) with 3 versions and CMIP5. The mechanism by which meteorological characteristics and geographical matrices can drive air pollution based on monitoring data at the site-scale was also examined. We found the total global emission of major air pollutants increased 1.32 times during 1970–2010. Air pollutant emissions will increase 2.89% and 4.11% in China and developed countries when the scenario of only maximum technically feasible reductions is performed (V4a) during 2020–2050. However, it will decrease 19.33% and 6.78% respectively by taking the V5a climate scenario into consideration, and precipitation variation will contribute more to such change, especially in China. Locally, the air circulation mode that is dominated by local geographical matrices and meteorological characteristics jointly affect the dilution and diffusion of air pollutants. Therefore, natural conditions, such as climate changes, meteorological characteristics and topography, play an important role in spatial air pollutant emissions and fluctuations, and must be given more attention in the processes of air pollution control policy making.

**Keywords:** air pollutant; GAINS model; climate change; life expectancy loss

## 1. Introduction

Air pollution is a serious global environmental problem [1,2]. Emission of air pollutants and aggravation of the pollution have long-term negative consequences on human health and life [3–7]. From 1970 to 2010, the total emission of the 10 major global air pollutants showed a fluctuating upward trend. The total emission of air pollutants in 2010 was $1.1534 \times 10^6$ tons, which was 1.32 times that in 1970 (Figure 1b). Among all sources of air pollutant emissions, anthropogenic sources, represented by residential and other sectors, manufacturing industries and construction, and road transportation, are main sources of air pollutants. From 1970 to 2010, the above-mentioned human activities produced a total of $2.8532 \times 10^7$ tons of air pollutants, which was 65.88% of the total emission of the 10 major air pollutants. Meanwhile, natural pollution sources, such as wildfires, dust, natural biogenic and lightning emissions, are still non-negligible sources of air pollutant emissions [8–19]. Up to now, several studies have demonstrated that climate change has significant influence in natural sources of air pollutant emissions [16–20]. However, the available studies which have explored the impacts of climate change to air pollutant emissions have mostly focused on a certain kind of air pollutant or

a certain source of air pollutant emission. It is necessary to research the correlation between total air pollutant emissions and climate change.

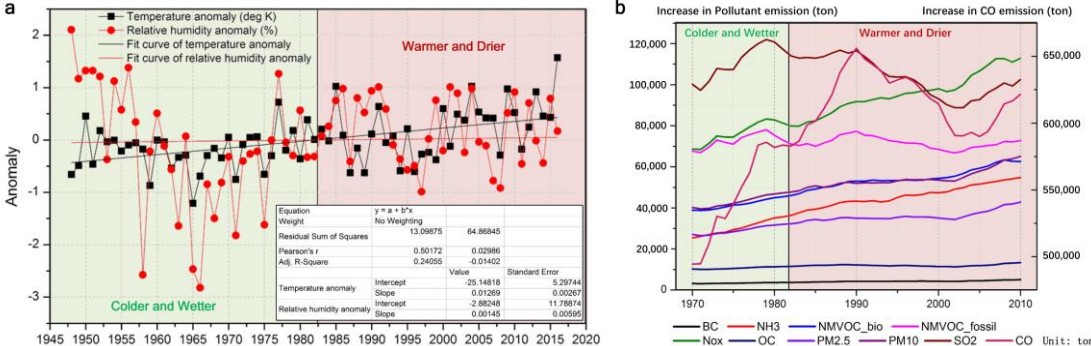

**Figure 1.** Short-term climate change trends and the variations of major air pollutant emissions. (**a**): Global annual average air temperature and relative humidity during 1948–2016. (**b**): Global increase in annual total emissions of ten major air pollutants from 1970 to 2010 (unit: ton). The right vertical axis indicates global increases in CO emissions, and the left vertical axis indicates global increases in the other 9 air pollutants' emissions. (Data source: The global climate data used in Figure 1a was downloaded from the National Oceanic and Atmospheric Administration (NOAA) website. The global air pollutant emission data used in Figure 1b was downloaded from the Emissions Database for Global Atmospheric Research website.)

Air pollutant variations have multiple relationships with not only human activities but also natural conditions including terrain, land cover, climatic characteristics, and local weather conditions [21,22]. Although human activities can considerably affect the emission, concentration and spatial distribution of air pollutants, atmospheric circulation and weather conditions are still the core dynamic forces of air pollutant distribution, and consequently, determine the dilution and diffusion of air pollutants [22–29]. Researchers have indicated that weather plays a crucial role in the occurrence of air pollutant episodes [19,20,30–32]; however, the studies which have explored the relationship between the change in emissions of all air pollutants and the climate change from a global perspective are not abundant. Since the air flow patterns and climate change characteristics largely depend on local geographical environments, studies on the effects of weather conditions on air pollution in different geographical zones are needed [33]. The existing studies on investigating how climate change and geographical conditions affect and influence the variations of air pollutants around the globe do not include a variety of terrain simultaneously in one study. Air pollution has been a worldwide environmental problem since the beginning of the industrial revolution. Therefore, studies performed to explore the dynamic mechanism of the spatio-temporal variations of air pollutants under different geographical conditions is urgently needed to improve the settled environments in which we live [1].

In this study, we simulated and analyzed the response of air pollutant emissions to climate change and its effects on human life expectancy loss by integrating the air pollutant emission simulation model (GAINS) with 3 versions and climate forecast model (CMIP5) data. The total amount of 10 air pollutants and greenhouse gases ($CO_2$, $CH_4$, BC, CO, $NH_3$, NOx, $N_2O$, PM, $SO_2$, VOC, etc.) in the future in developed countries (includes 43 countries of European, America, Australia) and 34 provinces of China (the largest developing country in the world) were simulated. Then, the response curves were given to explore how climate change affects air pollutant emissions. The details of this dynamic mechanism were explained at local scales under different climate background, terrain conditions and development levels. Based on the limitation of previous studies, the major goal of our research was to explore the correlation between total air pollutant emissions and climate change, and to focus on the dynamic mechanism of natural factors, especially for local weather conditions and local terrain conditions, and the combined effect of the spatial diffusion of various of air pollutants.

## 2. Materials and Methods

### 2.1. Data

The data used were due to availability. All the emission data of air pollutants used in this study were downloaded from the Emissions Database for Global Atmospheric Research website (http://edgar.jrc.ec.europa.eu/gallery.php?release=v431_v2&substance=SO2&sector=TOTALS). All the meteorological data in China used in this study were downloaded from the China meteorological data network (http://data.cma.cn/). All the global climate data (including CMIP5 data) used in this study were downloaded from the National Oceanic and Atmospheric Administration (NOAA) website (https://www.esrl.noaa.gov/psd/thredds/catalog/Datasets/ncep.reanalysis/other_gauss/catalog.html) [34]. Other relevant data in this study are available in our supplementary file.

### 2.2. The GAINS Model

The emissions of air pollutants are significantly correlated with atmospheric environment, geographical conditions and human activities, etc. Therefore, not only natural factors, but also series of policies and legislations should be taken into consideration in the process of simulating air pollutant emissions.

The GAINS model is developed to perform a simulation of air pollution change in the future under different scenarios with a rigorous consideration of human activities in order to improve air quality. Moreover, it also provides policy exploration to control local air pollution and climate warming caused by greenhouse gases at all scales around the world. Currently, 10 air pollutants and greenhouse gases ($CO_2$, $CH_4$, BC, CO, $NH_3$, NOx, $N_2O$, PM, $SO_2$, VOC, etc.) in 165 regions can be simulated by using 3 versions of the GAINS model, including V4a, V5, V5a (Table 1). The database that the GAINS model uses in the process of air pollutant emission simulation is mostly based on international energy and industrial statistics, emission inventories and data supplied by related departments of countries. Studies have proven that GAINS can contribute more to studies on air pollutant emissions and global climate change at different scales [35,36].

**Table 1.** Scenarios designed for air pollutant emission simulation (GAINS) model [37]. (http://www.iiasa.ac.at/web/home/research/researchPrograms/air/Global_emissions.html).

| Version | Release Date | Period Covered | Scenarios |
|---|---|---|---|
| V3 | November 2013 | 2005, 2008, 2009, 2010 | No future scenarios were developed |
| V4a | January 2014 | 2005, 2010, 2030, 2050 | Reference (assuming current legislation for air pollution—CLE). Maximum technically feasible reductions (MTFR) |
| V5 | April 2014 | 1990–2030, 2040, 2050 | Reference (assuming current legislation for air pollution—CLE), No further control (NFC). Short lived climate pollutants mitigation (SLCP) |
| V5a | July 2015 | 1990–2030, 2040, 2050 | Reference (assuming current legislation for air pollution—CLE), Short lived climate pollutants mitigation (SLCP), Maximum technically feasible reductions (MTFR), Climate scenario (2 degrees, CLE) |
| V6 | Forthcoming | | |

The GAINS model was designed to track the air pollutants based on various of economic activities, such as energy consummation, industrial production and agriculture [37]. These economic activities were the main driving forces of air pollutant emissions which were selected as the basis of the emission scenarios setting. Different from other air pollution simulation models, the characteristics of specific regions and sources of air pollutants were considered in air pollution evaluation. Then, technical and non-technical measures were applied to evaluate the potential cost of air pollutant emission reduction,

simulate the emission accumulation and dispersion process and calculate the impact indicators of air pollutant emissions on human health. Depending on the correlation between the indicators that reflect the impacts of air pollutant emissions on human life expectancy and PM2.5, the human life expectancy loss in the future can be simulated under different scenarios using the GAINS model.

We performed 3 versions of the GAINS model (V4a, V5, V5a) to explore the changes of total emissions of 9 air pollutants in 43 countries in the world and 33 provinces of China (Hong Kong and Macau were merged as one administrative area of China), and to calculate the subsequent impacts of PM2.5 on human life expectancy loss in the future. The 43 countries are all developed countries with relatively sound public policy measures and awareness aimed at improving air quality. China is known as the largest developing country in the world and characterized by an imbalanced development from eastern to western. Moreover, not only the developed areas of China but also the developing regions are all facing the challenge of air pollution during the past decades. Since developing countries are still in the process of large-scale consumption of traditional energy and only the initial stage of air pollution abatement, we suppose that air pollution will keep decreasing in developed countries and increase in developing countries at a middle stage of development. We have assessed the total air pollutant emissions of countries with different socio-economic development levels and explored their potential correlation with climate change in the future. More details about the performing of the GAINS model are displayed in the 'GAINS Online: Tutorial for advanced users' provided by the International Institute for Applied Systems Analysis (http://www.iiasa.ac.at/web/home/research/researchPrograms/air/GAINS-tutorial.pdf).

### 2.3. Spatial Correlation Analysis Method

The study first assumes a null hypothesis that climate change factors do not show a statistical correlation with air pollutant emissions. Then, point extraction from the spatial statistical analysis module of ArcGIS 10.2 was carried out to obtain all data from climate change factor layers and air pollutant emission layers. At the same time, in consideration of all potential spatial correlations between climate change and air pollutant emission changes in all geospatial environments, this study extracted all pixel values in every raster layer (each layer contains a total of $3600 \times 1825$ raster pixels), thus achieving a global spatial correlation analysis of climate change and air pollutant emissions.

If $p < 0.05$ was obtained for the test results, then the null hypothesis would be rejected, i.e., a significant statistical correlation exists between climate change factors and air pollutant emission factors. If $p > 0.05$ was obtained for the test results, then the null hypothesis would be accepted, i.e., no significant statistical correlation exists between climate change factors and air pollutant emission factors.

### 2.4. Terrain Relief Model

Terrain relief refers to the difference between maximum and minimum elevations within a given area [38]. It is a macroscopic marker that describes the terrain characteristics of a region. In view of this, this study proceeded from the concept of relief and used the product of altitude difference and proportion of flat ground to measure relief. The Geomorphology Survey and Mapping Committee International Geographical Union classified the slope of landforms into the following: $0°{\sim}0.5°$ as plains, $>0.5°{\sim}2°$ as a gentle slope, $>2°{\sim}5°$ as a moderate slope, $>5°{\sim}15°$ as a steep slope $>15°{\sim}35°$ as a very steep slope, $>35°{\sim}55°$ as a cliff, and $>55°{\sim}90°$ as a vertical wall. Based on this, this study defined terrain with gradient lower than $2°$ as plains. Additionally, we combined the measurement method used by Feng et al. in terrain relief in China, corresponding with a data resolution rate of $0.1° \times 0.1°$ in the study [39], and used 10 km $\times$ 10 km as a basic measurement unit to construct a global relief measurement model based on the concept of relief:

$$\text{RDLS} = \{[\text{Max}(h) - \text{Min}(h)] \times [1 - (P(A))/A]\}/500$$

In the equation, RDLS is the relief of the terrain, Max(h) and Min(h) are the maximum and minimum elevations of a 10 km × 10 km region, respectively; P(A) is the area of flat land in the 10 km × 10 km region; A is the total area of the region; and 500 is the datum mountain height (unit: m).

*2.5. Redundancy Analysis Model*

Spatial differentiation characteristics exist in the geographical distribution of meteorological factors, causing the dynamic factors that affect pollutant concentration to show different gradient structure distributions. An example of this is how temperature affects gas expansion on a local scale. Only when elements with spatial attributes are included in the analysis of driving mechanisms for geographical objects to examine air pollutant variations under geospatial differentiation patterns, it is possible to accurately describe the dynamic mechanisms of local climate conditions on air pollutant concentrations in that area. Therefore, this study introduces the concept from ecology where plant communities are arranged according to gradients of environmental factors: the concentration of a few major air pollutants (CO, $NO_2$, $SO_2$, $O_3$, PM2.5, and PM10) were viewed as "species" (dependent variable) and meteorological markers that can reflect local climate characteristics and extreme weather were viewed as "environmental factors" (independent variable) to further explain the relationship between the air pollutant concentration and environmental factors and to demonstrate the driving mechanisms of environmental factors on the dilution and diffusion of air pollutants.

According to the relationship between species and environmental factors, the gradient analysis model can be divided into the linear model and the unimodal model. As the name suggests, a linear model means that a linear correlation exists between model variables (model variables include "species" and "environmental" factors). The unimodal model refers to positive directional changes in species when environment factors change until a certain threshold value is reached, where it becomes negative directional change [40]. In the model, the relationship in input variable function is the unimodal curve. In addition, according to the type of variables entered into the model, the gradient analysis model can be further divided into an indirect ordination model (model variable only contains "species") and direct ordination models (model variables include "species" and "environmental factors"). Model construction covers four aspects, namely driving hypothesis analysis and factor screening → driving scenario and model design → driving factor processing → model selection. The gradient analysis results are mainly based on the two-dimensional ordination diagram between species and environmental factors. If the angle between species and environmental factors is an acute angle (<90°), this shows that both variables have a positive correlation and the smaller the angle, the greater the correlation. Conversely, if the angle is an obtuse angle (>90°), or the acute angle is formed with the extension in the reverse direction, this shows that both variables have a negative correlation, and the larger the angle, the greater the correlation. In addition, the length of the ordination line between species and factors can also show the degree of correlation between the two, and the length of the ordination line is directly proportional to the degree of correlation.

As multiple function relationships exist between the species and environmental factors, the correlation is not purely linear or unimodal. Therefore, the most optimal gradient analysis model should be selected according to the corresponding relationship presented between the species and the environment, in order to explain the effects of environmental changes on species. Currently, the selection method for widely used gradient analysis models is mostly based on detrended correspondence analysis (DCA) analysis results of species data. In four ordination axes, if the first axis is longer than 4, selection of the unimodal model ordination (correspondence analysis (CA), canonical correspondence analysis (CCA), DCA) is more appropriate. If it is less than 3, the linear models (PCA, redundancy analysis model (RDA)) are more reasonable. If it is between 3 and 4, both unimodal and linear models are suitable. Accordingly, this study selected the gradient analysis model with a total of 66 groups of variables in nine cities with six types of geographical matrix. According to the analysis results of Supplementary Table S3, in the 66 groups, the first ordination axis of the driving mechanisms of air pollutant dilution and diffusion is less than 3. Therefore, the linear gradient analysis model is

more superior to the unimodal gradient analysis model. In addition, this study aims to examine the driving mechanisms of regional meteorological conditions on the dilution and diffusion of air pollutants through gradient analysis. The model variables include both dependent and independent variables, which belong to classical indirect ordination. Therefore, the redundancy analysis model (RDA) is more suitable.

### 2.6. Spatial Dynamic Model

Spatial dynamics represent the degree of change of a certain geographical phenomenon within a unit of spatio-temporal range [41]. The specific physical meaning can be expressed as the degree of change at the end of the study period relative to the initial year.

$$D = \frac{a_j - a_i}{a_i}$$

In the equation, $a_i$ is the spatial distribution status of air pollutants in the initial year; $a_j$ is the spatial distribution status of air pollutants at end of the study period.

### 3. Results

#### 3.1. Dynamic Change of Global Air Pollutant Emissions

There are significant differences in the spatial distribution of the average annual emissions of various air pollutants [42,43] (Supplementary Figures S2–S11). Overall, countries with large populations and energy consumption are regions with high emissions of various pollutants [44–46]. To further analyze changes in the spatial distribution of global air pollutant emissions, we propose a spatial distribution dynamic model for the analysis of spatial variation in the annual emissions of each air pollutant and then extract the spatial distribution of highly dynamic regions (Figure 2). Greenland, in the Arctic Circle, the surrounding area of Hudson Bay in Canada, Asia (excluding North Asia), the African continent, the Brazilian highlands, and the Pampas in South America were the regions where high spatial distribution dynamics of air pollutant emissions were concentrated.

Spatial overlap was carried out between the abovementioned regions with concentrated distributions of various types of air pollutants and the spatial distribution of air pollutant source areas such as global traffic (roads, railways, and aviation), human settlements, and farmland (Supplementary Figures S12 and S13). However, according to the spatial variation characteristics of annual average emission dynamics of various air pollutants in 1970–2010, it is difficult to accurately explain the distribution variation in spatial dynamics of air pollutant emissions with human activities, which are the major source of air pollutant emissions. In some regions with high spatial dynamics of air pollutant emissions, such as the inner Arctic Circle and the African rainforest region, there are few footprints of human activities. However, these regions are sensitive to fluctuations in natural conditions, such as climate change (Supplementary Figures S14 and S15), indicating that changes in the natural environment are the core dynamic conditions driving the high spatial dynamics of air pollutant emissions.

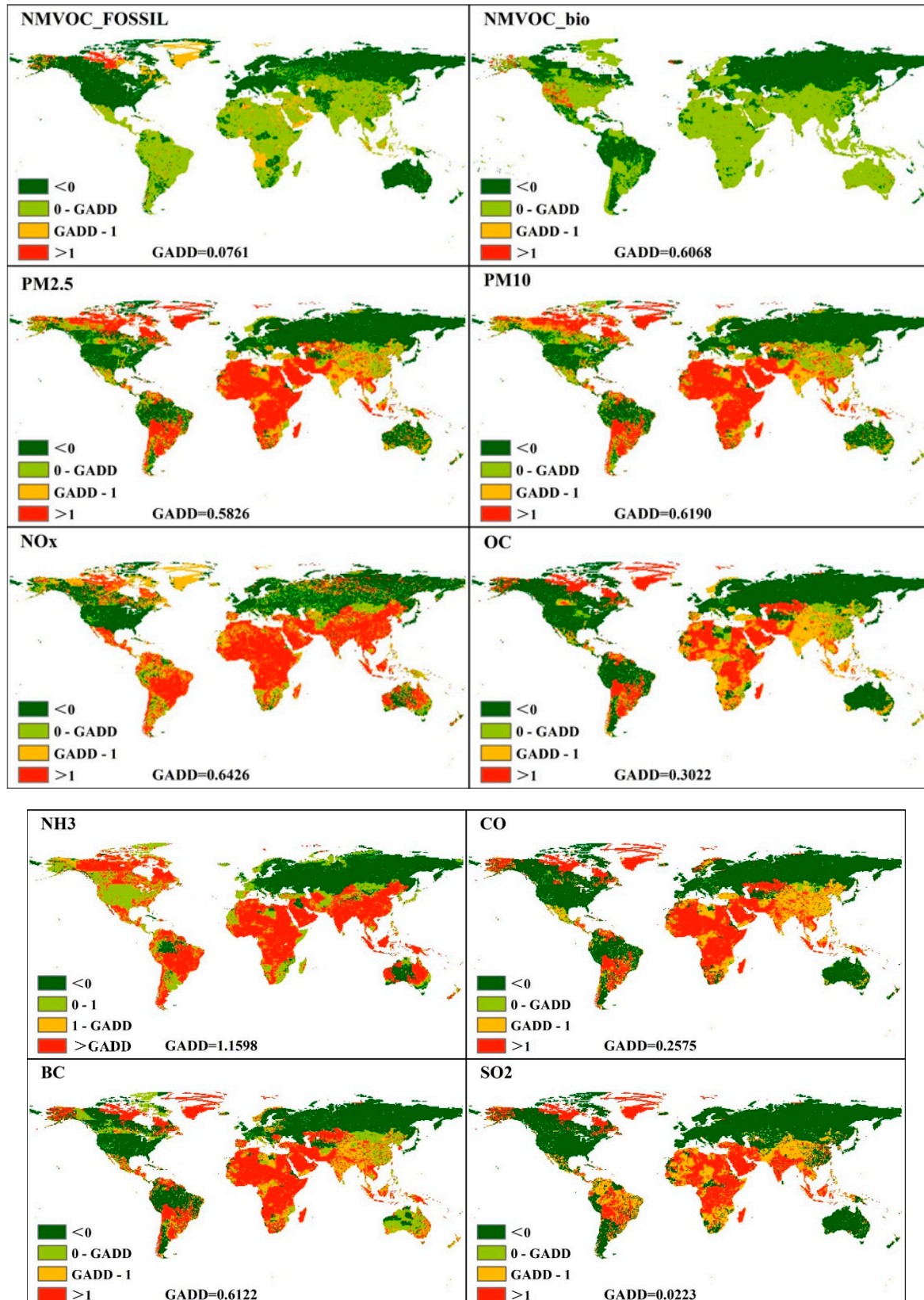

**Figure 2.** Changes of the spatially dynamic degree of major air pollutant emissions during 1970–2010. The definition of spatial dynamic degree is explained in the methods section. Global Average Dynamic Degree (GADD) refers to the average value of all the units' spatial dynamic degree in each sub-graph.

### 3.2. Spatial Correlation between Air Pollutant and Natural Conditions

In order to investigate the effects of natural conditions on spatial distribution changes in air pollutants, we selected terrain relief, temperature dynamics, precipitation dynamics, air pressure dynamics, relative humidity dynamics, and wind speed dynamics as natural condition indicators. Taking into account the potential spatial correlation between climate change and changes of air pollutant distributions in all geospatial environments, all pixel values in each raster layer were extracted (each layer contains $3600 \times 1825$ raster pixels), thus achieving global spatial correlation analysis of climate change and air pollutants. Table 2 shows that temperature and relative humidity are the core meteorological dynamic factors driving variations in the spatial dynamics of various pollutants ($p < 0.05$). Particularly for PM 2.5, PM10, NMVOC_bio, and $NH_3$, climate warming and drying due to combined changes in temperature and relative humidity and its spatial dynamics showed strong spatial correlation ($p < 0.01$). Moreover, changes in the spatial dynamics of air pollutants also showed some correlation with terrain relief, precipitation dynamics, air pressure dynamics, and wind speed dynamics. This shows that, even though the major source of air pollutants is human activities, global climate warming and drying still strongly affects the spatial distribution and diffusion of air pollutants. Additionally, the geographical matrix, such as terrain relief, can also limit the effects of atmospheric circulation on transport power intensity and the path of air pollutants locally.

**Table 2.** Spatial correlation analysis between global air pollutants and related natural indicators. The natural indicators from first to sixth are relief amplitude, air temperature, precipitation, air pressure, relative humidity, and wind speed respectively. Indicators from seventh to sixteenth are $SO_2$, PM2.5, PM10, OC, NOx, NMVOC_FOSSIL, NMVOC_bio, $NH_3$, CO, and BC respectively. $p < 0.05$ refers to a significant correlation.

| Indicators | 1 | 2 | 3 | 4 | 5 | 6 | 7 | 8 | 9 | 10 | 11 | 12 | 13 | 14 | 15 | 16 |
|---|---|---|---|---|---|---|---|---|---|---|---|---|---|---|---|---|
| 1 | 1 | −0.0195 | −0.0584 | 0.0447 | −0.0278 | −0.0313 | −0.0883 | −0.0078 | −0.0319 | −0.1437 | −0.0605 | −0.2086 | −0.0118 | 0.0109 | −0.3251 | −0.3376 |
| 2 | −0.0195 | 1 | 0.4687 | −0.1938 | 0.0998 | 0.0428 | −0.0320 | −0.0016 | −0.0083 | −0.0421 | −0.0221 | −0.0704 | −0.0080 | 0.0000 | −0.1159 | −0.1207 |
| 3 | −0.0584 | 0.4687 | 1 | −0.1338 | 0.2099 | −0.0200 | −0.1081 | −0.0100 | −0.0425 | −0.1718 | −0.0747 | −0.2615 | −0.0143 | −0.0009 | −0.4029 | −0.4171 |
| 4 | 0.0447 | −0.1938 | −0.1338 | 1 | −0.0125 | −0.0174 | 0.0825 | 0.0086 | 0.0336 | 0.1291 | 0.0563 | 0.1988 | 0.0118 | 0.0011 | 0.3021 | 0.3137 |
| 5 | −0.0278 | 0.0998 | 0.2099 | −0.0125 | 1 | 0.0084 | −0.0514 | −0.0041 | −0.0062 | −0.0841 | −0.0354 | −0.1297 | −0.0069 | −0.0007 | −0.1975 | −0.1982 |
| 6 | −0.0313 | 0.0428 | −0.0200 | −0.0174 | 0.0084 | 1 | −0.0930 | −0.0097 | −0.0367 | −0.1628 | −0.0616 | −0.2260 | −0.0108 | −0.0005 | −0.3619 | −0.3745 |
| 7 | −0.0883 | −0.0320 | −0.1081 | 0.0825 | −0.0514 | −0.0930 | 1 | −0.0093 | 0.1284 | −0.1220 | 0.9896 | −0.3000 | −0.0155 | −0.0003 | −0.4354 | −0.4646 |
| 8 | −0.0078 | −0.0016 | −0.0100 | 0.0086 | −0.0041 | −0.0097 | −0.0093 | 1 | 0.0343 | −0.0219 | −0.0029 | −0.0307 | −0.0016 | 0.0000 | −0.0485 | −0.0500 |
| 9 | −0.0319 | −0.0083 | −0.0425 | 0.0336 | −0.0062 | −0.0367 | 0.1284 | 0.0343 | 1 | −0.0860 | 0.1486 | −0.1145 | −0.0062 | −0.0001 | −0.1942 | −0.2007 |
| 10 | −0.1437 | −0.0421 | −0.1718 | 0.1291 | −0.0841 | −0.1628 | −0.1220 | −0.0219 | −0.0860 | 1 | −0.1465 | −0.4698 | −0.0259 | −0.0005 | −0.1549 | −0.3376 |
| 11 | −0.0605 | −0.0221 | −0.0747 | 0.0563 | −0.0354 | −0.0616 | 0.9896 | −0.0029 | 0.1486 | −0.1465 | 1 | −0.2064 | −0.0106 | −0.0002 | −0.3206 | −0.3344 |
| 12 | −0.2086 | −0.0704 | −0.2615 | 0.1988 | −0.1297 | −0.2260 | −0.3000 | −0.0307 | −0.1145 | −0.4698 | −0.2064 | 1 | −0.0364 | −0.0008 | −1.0390 | −1.0910 |
| 13 | −0.0118 | −0.0080 | −0.0143 | 0.0118 | −0.0069 | −0.0108 | −0.0155 | −0.0016 | −0.0062 | −0.0259 | −0.0106 | −0.0364 | 1 | −0.0002 | −0.0573 | −0.0591 |
| 14 | 0.0109 | 0.0000 | −0.0009 | 0.0011 | −0.0007 | −0.0005 | −0.0003 | 0.0000 | −0.0001 | −0.0005 | −0.0002 | −0.0008 | −0.0002 | 1 | −0.0012 | −0.0012 |
| 15 | −0.3251 | −0.1159 | −0.4029 | 0.3021 | −0.1975 | −0.3619 | −0.4354 | −0.0485 | −0.2007 | −0.1549 | −0.3206 | −1.0390 | −0.0573 | −0.0012 | 1 | −1.2398 |
| 16 | −0.3376 | −0.1207 | −0.4171 | 0.3137 | −0.1982 | −0.3745 | −0.4646 | −0.0500 | −0.2007 | −0.3376 | −0.3344 | −1.0910 | −0.0591 | −0.0012 | −1.2398 | 1 |

Significantly correlated: ▢ $P < 0.05$ ▢ $P < 0.01$

As there are significant scale differences and spatial heterogeneity characteristics in the global geographical environment, the spatial dynamic mechanisms of air pollutants on a local scale are more diverse than the macroscopic mechanisms on a global scale [47–49]. As local-scale variations in weather conditions between different regions are not the same against the background of global climate change, we proceeded from macroscopic research results on a global scale to carry out in-depth extractions of effector mechanisms on how natural conditions can affect the transmission and diffusion of air pollutants on a local scale.

### 3.3. Response of Air Pollutant Emissions to Climate Change

Spatial correlation analysis showed that spatial air pollutant fluctuations negatively correlated with global warming and drying due to changes in global temperatures and relative humidity. In order to explore the response relationship of air pollutant emissions to climate change, we firstly simulated the change of air pollutant emissions under a warming and dry climate change background in the future (Supplementary Figure S16). According to the simulation result of 33 provinces in China and 43 developed countries including European countries, American, Australia, Canada and New Zealand, etc., (supplementary file) the total emissions of air pollutants will increase 2.89%

and 4.11% in China and developed countries respectively in V4a without the consideration of climate change (Supplementary Tables S1 and S2). On the contrary, the total air pollutant emissions in China and developed countries will decrease by 19.33% and 6.78% in V5a by taking climate change into consideration (Supplementary Tables S1 and S2). This indicates that climate change characterized with warming and dryness will result in a decreasing of total emissions of air pollutants in the future. Especially for China, the largest developing country in the world, the decrease of precipitation will lead to an increase of air pollutant emissions, such as CO, PM2.5, PM10, BC, in V5a (Supplementary Figure S22). If no further control is taken, the response of the emission increase of PM2.5, PM10 and BC to precipitation decrease seems more significant in V5 (Supplementary Figure S21). Otherwise, only a minor relationship can be found between air pollutant emissions and climate change in V4a (Supplementary Figure S20). However, in neither V5a, V5 nor the V4a, are the responses of air pollutant emissions to climate change in developed countries not so notable generally (Supplementary Figures S17–S19). This is mainly attributed to the completed public facilities and public awareness for preventing air pollutant emissions in developed countries.

*3.4. Potential Effects of Air Pollutant Emissions Because of Climate Change on Human Health*

Depending on the GAINS model and taking the effects of PM2.5 on human life expectancy loss in east Asia as a case study, we further explored the potential effects of air pollutant emissions on human health against the background of climate change in the future. Our findings prove that air pollution under the V5a climate scenario (CLE) will contribute more to a low life expectancy loss than any other scenarios (Figure 3p–r). The scenario with short lived climate pollutant mitigation also indicates a lower life expectancy loss, which is generally smaller than 100 months (Figure 3m–o). On the contrary, the scenario of no further control (V5_NFC) and the baseline scenario without policy change for air pollutant emissions will lead to a higher life expectancy loss. Especially for places with a basin terrain, the life expectancy loss in scenarios of baseline and V5_NFC will be above 100 months in 2020, 2025 and 2030. According to the analysis in Section 4.1, basin cities easily suffer heavy air pollution because of their blocked terrain conditions. Consequently, air pollution in regions with blocked terrain will more probably cause a higher life expectancy loss.

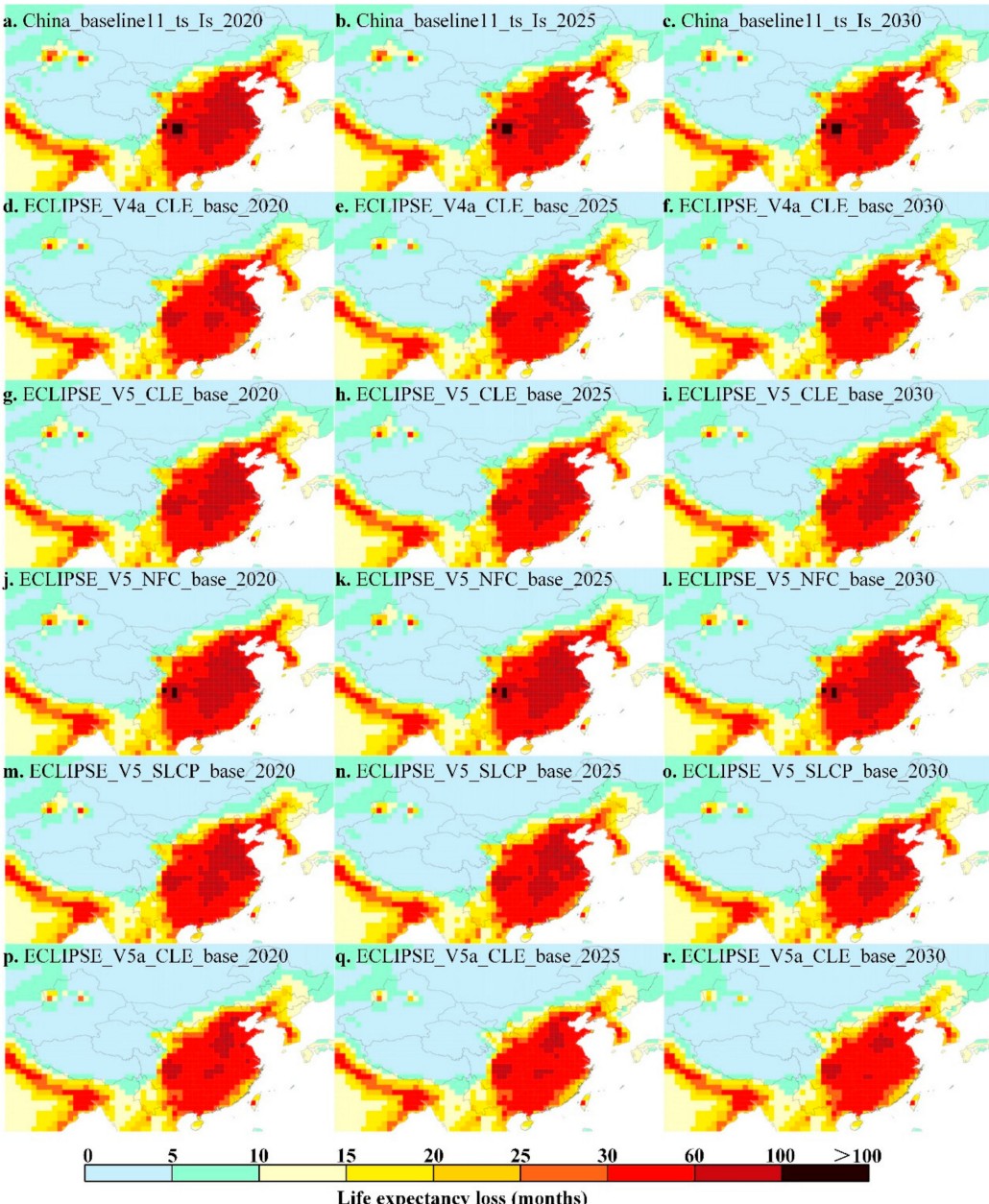

**Figure 3.** The simulation results of the effects of PM2.5 on human life expectancy loss in the future under different scenarios (Baseline, V4a, V5, V5a). (**a**,**d**,**g**,**j**,**m**,**p**) show the simulation results under different scenarios by 2020. (**b**,**e**,**h**,**k**,**n**,**q**) show the simulation results under different scenarios by 2025. (**c**,**f**,**I**,**l**,**o**,**r**) show the simulation results under different scenarios by 2030.

Furthermore, all high life expectancy loss under different scenarios (above 30 months) will take place in developed areas with crowded population density. For instance, the high life expectancy loss caused by air pollutant emissions in China are all concentrated in the eastern areas of the Chinese population line (Heihe–Tengchong line), which is famous for the highest population density all over the world (Figure 3). It needs to be further pointed out that all life expectancy loss is positively related with the economy growth rate. Not only eastern China, but also the river Ganges in northern India, Ganges Delta of Bangladesh and the west coast of South Korea, are all characterized with a high economy growth rate in the past or present. Especially for China and India, the first and second largest developing countries in the world, which are experiencing and continuing a higher speed of economy growth. However, such increase comes at the expense of environment pollution and human

health [50–52]. Irrational urbanization processes and ignorance of environment protection in policy all for economic development make air pollutant emissions the most serious environment problem that endanger health in developing countries [53–55].

Generally speaking, as the predictions of the GAINS model show, air pollutant emissions will continually lead to severe harm to human health, if there is no policy control for climate change in the future. Although a series of measures have been produced to cope with air pollution, the baseline scenario under the current air pollution controlling methods still show a high life expectancy loss (Figure 3a–c). How to address the health burden raised by air pollutant emissions in a changing climate background has been a long-term task for China and related countries with serious air pollution problems [50].

## 4. Discussion

### 4.1. Mechanism of Air Pollutant Diffusion

Local air circulation is the basis for driving spatial dynamic variations of air pollutants locally [56], and the geographical matrix determines the local air circulation model. Local atmospheric circulations are formed due to complicated geography matrices, such as heterogeneous underlying surface, terrain fluctuation, and different land cover types [57–59]. Among these, terrain restricts the dynamic and thermal basis for the formation of atmospheric circulation, thereby limiting, increasing, or modifying the direction and intensity of air circulation [58]. Meteorological indicators, such as wind, temperature, and air pressure, can influence the formation, paths, and intensity of air flow directly or indirectly, which plays an extremely important role in the change of concentrations and diffusion paths of air pollutants at the local scale [60–62]. In particular, few studies have explored the responding relationship between air pollutants and weather conditions from the perspective of local atmospheric circulations under complicated geography matrices on a local scale [33,63]. We considered nine cities (Supplementary Figure S23) that belong to six types of different geographical matrices (river valley cities, cities on plains, mountain–plain transitional cities, coastal cities, cities with karst topography, and basin cities) as examples and analyzed the effects of local air circulation and local meteorological conditions on the formation of air pollutants based on actual measurement data pertaining to concentrations of a few major air pollutants and the actual data from meteorological indicators (Supplementary Figures S24–S26).

For river valley cities, the terrain is blocked, especially in cities where the direction of the river valley is not the same as the prevailing wind direction; therefore, the effects of the prevailing wind direction on local air circulation are insignificant. The latter is more dependent on the air pressure differences produced by temperature differences during the day and at night between the mountainous region and the river valley (Figure 4a,b). During the day, the mountainous region is heated faster than the river valley and the air flows from the mountainous region to the river valley. At night, the temperature of the mountainous regions decreases faster than the river valley and air flows from the river valley to the mountainous region. This results in the local air circulation model of river valley cities, which presents itself as an almost vertical bidirectional circulation between the mountainous region and the river valley during the day and at night. RDA (Redundancy analysis) analysis results show that in river valley cities, the air pollutant concentration shows a significant negative correlation with temperature, precipitation, and wind speed (Supplementary Figures S27 and S28; Supplementary Tables S4 and S5). Due to the sealed terrain conditions of the river valley region, this results in local background climatic conditions having less impact and the effects of weather on the transport of air pollutants being more dependent on the intensity of mountain–river valley air circulation. This is specifically characterized as changes in wind conditions due to temperature, precipitation, and differences in horizontal pressure gradient. Therefore, with regard to river valley cities, the circulation model as a result of terrain constraints, determined that it is difficult for air

pollutants to diffuse into other regions. In addition, weal wind intensity and frequent temperature inversion also further increased the difficulty of air pollutant diffusion.

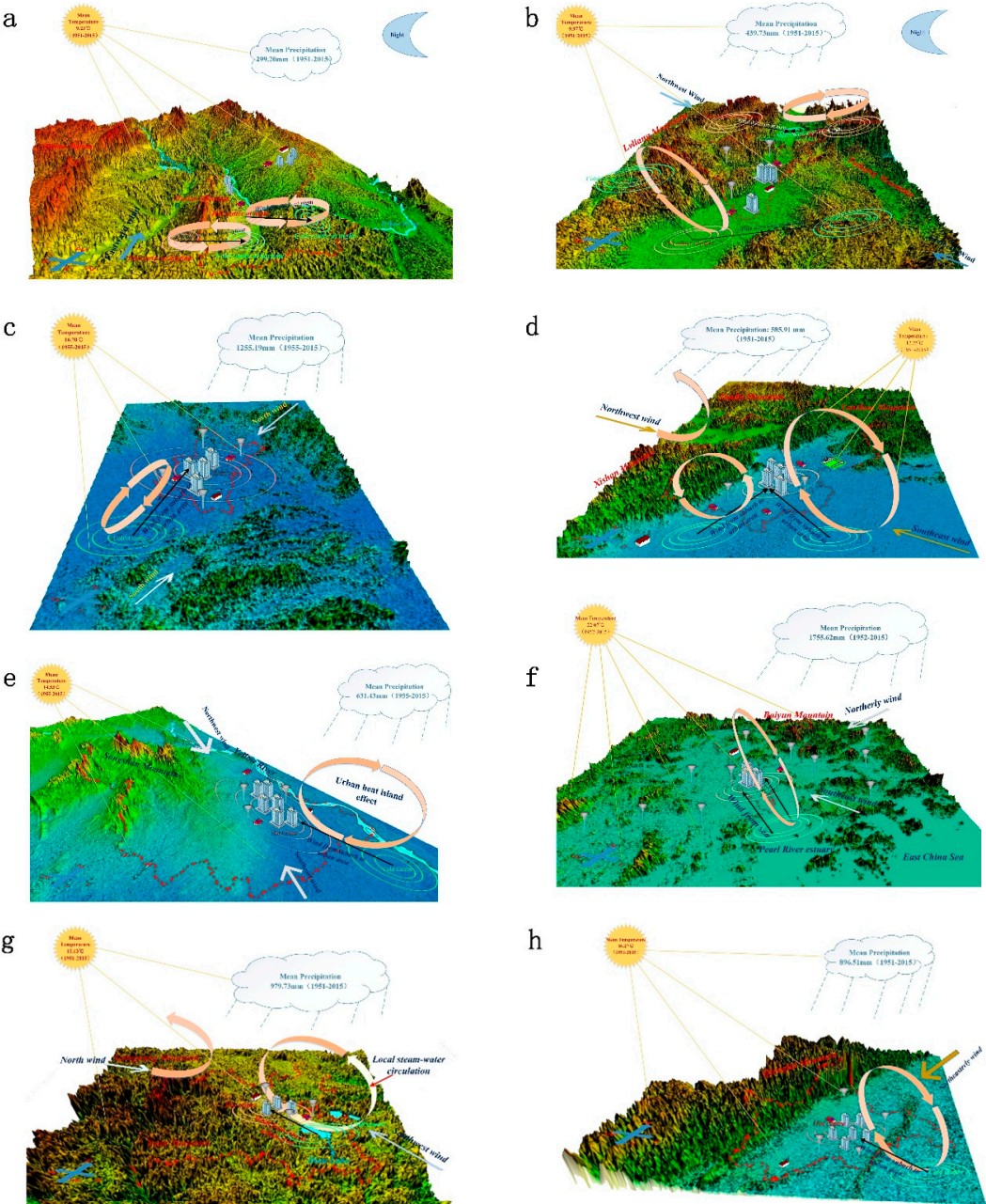

**Figure 4.** The air circulation model under complicated geographic and meteorological conditions. (**a–h**) show the air circulation model of 8 cities with different geographic and meteorological conditions. Among the 8 cities displayed in the Figure, Lanzhou and Taiyuan in China represent river valley cities, Wuhan represents a plain city, Beijing and Zhengzhou represent transition cities from mountains to plains, Guangzhou represents a coastal city, Kunming represents a city in a karst landform, and Chengdu represents a basin city.

In cities on plains, because the terrain is flat and open, the prevailing wind direction can directly affect the diffusion of air pollutants; therefore, wind speed plays a key role in the diffusion of air pollutants in this type of city. This is especially true for PM2.5, CO, $SO_2$, and $NO_2$, where wind speed shows a significant negative correlation with the concentrations of these air pollutants (Supplementary Figure S29; Supplementary Tables S4 and S5). However, temperature differences

between built-up areas and rural areas caused by the urban heat island effect causes increased differences in the horizontal air pressure gradient between urban and rural areas (Figure 4c). This leads to the formation of air circulation from rural areas to the city and blocks the diffusion of urban air pollutants, to some extent. Therefore, a significant positive correlation between air pressure and air pollutant concentration is formed.

In mountain–plain transition cities, tall mountains weaken the driving effect of prevailing winds on the spread of urban air pollutants, to some extent. In particular, in cities where the orientation of the mountain is perpendicular to the direction of the prevailing wind, the intensity of the prevailing wind and concentration of air pollutants in the city show a strong negative correlation (Supplementary Figures S30 and S31; Supplementary Tables S4 and S5). At the same time, the urban heat island effect still exists. Local urban–rural circulation created by differences in the horizontal air pressure gradient results in stagnation of air pollutants in low altitude of the city (Figure 4d,e). Ultimately, air pollutants cannot diffuse to other regions.

The climate characteristics of coastal cities mostly present as significant sea–land thermal differences. Consequently, the effects of sea–land air pressure differences on local air circulation are stronger than those of the prevailing wind direction and urban heat island effect (Figure 4f). The sea–land circulation determines the air pollutant movement model in coastal cities and determines that the core meteorological factor driving the spread of air pollutants in coastal cities is wind speed. Using Guangzhou and Shanghai in China as examples, we found that nearly all air pollutants show a strong negative correlation with wind speed (Supplementary Figures S32 and S33; Supplementary Tables S4 and S5).

In cities with karst topography, the presence of multiple unique peaks as a terrain characteristic increases surface roughness. These cities are mainly concentrated in small basins where the center is flat while the surroundings are highlands. For this reason, the main air circulation characteristic of this city is local circulation and the effects of large background air circulation (such as monsoon circulation) on the dispersion of air pollutants are weaker than those of the local circulation formed due to macroclimate and terrain characteristics (Figure 4g). As cities with karst topography are in a small, sealed terrain, this results in the circulation characteristics presenting as significant vertical atmospheric circulation. Therefore, the concentration of air pollutants increases or decreases due to the chemical and physical interactions among the atmosphere and precipitation. Using Kunming in Yunnan Province, China as an example of a classic city with karst topography, we found that the main urban air pollutants (PM2.5, PM10, $SO_2$, etc.) showed a significant negative correlation with precipitation (Supplementary Figure S34; Supplementary Tables S4 and S5).

As basin cities are surrounded by mountains and the terrain is blocked, the atmospheric transport is mostly internal transportation. Owing to this influence, the climate characteristics of basin cities mostly present as significant vertical climates and complex local microclimates. Outward atmospheric dispersion and transport ability are weak. The dispersion model for air pollutants is generally based on classic meteorological factors. In Chengdu (Figure 4h), China, the main air pollutants (PM2.5, PM10, CO, $SO_2$, $NO_2$, etc.) showed a negative correlation with precipitation, wind speed, etc. (Supplementary Figure S35; Supplementary Tables S4 and S5). This is consistent with the meteorological characteristics of this region: abundant rainfall, relatively less sunshine, complex internal circulation, and a diverse and complicated local microclimate.

## 5. Conclusions

This study explored the correlation between total air pollutant emissions to climate change worldwide, and took nine cities with typical terrain to analyze the mechanism by which meteorological characteristics and geographical matrices can drive air pollutants at the site-scale. It has been found that natural conditions, such as climate changes, meteorological characteristics, and topography, play important roles in the spatial diffusions of air pollutants. This study also simulated the potential

effects of air pollutant emissions on human health by integrating the air pollutant emission simulation model (GAINS) and CMIP5.

In summary, short-term climate warming and drying has significant effects on, and correlation with, spatial fluctuations in the global distribution of major air pollutants. Moreover, air pollutant emissions show a significant response to the climate change in China, but little correlation in developed countries. At a local scale, the dilution and diffusion of air pollutants in a specific area is driven by the thermal and dynamic effects of the local circulation model, which is affected by geographical conditions (such as terrain relief and type of landforms, etc.) and comprehensive meteorological conditions.

Based on the research, we suggest that if no further controls are implemented, or further attention is paid to climate change and air pollutant emissions, then there will be potential threats to human health. Therefore, governments should adopt corresponding policies and measures to slow down the trend of global climate warming and drying, and pay more attention to air pollution control policy making.

**Supplementary Materials:** The following are available online at http://www.mdpi.com/2071-1050/11/13/3670/s1, Figure S1: Changes of the spatially dynamic degree of major air pollutants during 1970–2010, Figure S2: Average emission of global NMVOC_FOSSIL for ten years, Figure S3: Average emission of global NMVOC_bio for ten years, Figure S4: Average emission of global PM2.5 for ten years, Figure S5: Average emission of global PM10 for ten years, Figure S6: Average emission of global NOx for ten years, Figure S7: Average emission of global OC for ten years, Figure S8: Average emission of global $NH_3$ for ten years, Figure S9: Average emission of global CO for ten years, Figure S10: Average emission of global BC for ten years, Figure S11: Average emission of global SO2 for ten years, Figure S12: Global changes in transportation (road, railway and airport) and terrain relief, Figure S13: Global changes in land use and land cover, Figure S14: Global climatic conditions, Figure S15: Spatial heterogeneity of global wind direction, Figure S16: Global change of air temperature (**a**) and precipitation (**b**) during 2020–2050, Figure S17: Linear fitting of air pollutant emissions simulation and climate change during 2020–2050 under scenario V4a, Figure S18: Linear fitting of air pollutant emissions simulation and climate change during 2020–2050 under scenario V5, Figure S19: Linear fitting of air pollutant emissions simulation and climate change during 2020–2050 under scenario V5a, Figure S20: Linear fitting of air pollutant emissions simulation and climate change during 2020–2050 under scenario V4a, Figure S21: Linear fitting of air pollutant emissions simulation and climate change during 2020–2050 under scenario V5, Figure S22: Linear fitting of air pollutant emissions simulation and climate change during 2020–2050 under scenario V5a, Figure S23: Air pollution monitoring stations and meteorological monitoring stations in China, Figure S24: A series of long-term observations of meteorological conditions and air pollutants in eastern China, Figure S25: A series of long-term observations of meteorological conditions and air pollutants in central China, Figure S26: A series of long-term observations of meteorological conditions and air pollutants in western China, Figure S27: Diagram of redundancy analysis ordination of Beijing, Figure S28: Diagram of redundancy analysis ordination of Shanghai, Figure S29: Diagram of redundancy analysis ordination of Guangzhou, Figure S30: Diagram of redundancy analysis ordination of Taiyuan, Figure S31: Diagram of redundancy analysis ordination of Zhengzhou, Figure S32: Diagram of redundancy analysis ordination of Wuhan, Figure S33: Diagram of redundancy analysis ordination of Lanzhou, Figure S34: Diagram of redundancy analysis ordination of Chengdu, Figure S35: Diagram of redundancy analysis ordination of Kunming, Table S1: Changes of total emissions of 9 air pollutants in 43 developed countries, Table S2: Changes of total emissions of 9 air pollutants in China, Table S3: Driving model selection, Table S4: Results of Redundancy Analysis between Air Pollution and Weather Conditions, Table S5: Correlation between air pollution and weather conditions based on the results of redundancy analysis ordination.

**Author Contributions:** Conceptualization, Q.C.; Methodology, Q.C.; Validation, Q.C.; Formal Analysis, Q.C.; Investigation, Q.C.; Data Curation, Q.C.; Writing—Original Draft Preparation, Q.C.; Writing—Review and Editing, Q.C., M.L., F.L. and H.T.; Visualization, Q.C. and H.T.; Supervision, M.L. and F.L.

**Funding:** This research was funded by the National Key Research and Development Plan (Grant No. 2017YFB0504205).

**Conflicts of Interest:** The authors declare no conflict of interest.

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
