# Peer review of "Response of Global Air Pollutant Emissions to Climate Change and Its Potential Effects on Human Life Expectancy Loss"

_sustainability, doi:10.3390/su11133670_

Round 1

Reviewer 1 Report

My main issue with this paper is its novelty. The paper states that "thus far there has been no study investigating how climate change and geographical environment affect and influence the variations of air pollutants around the globe" (Line 80-81), which is not true. For example:

West, J. J., Smith, S. J., Silva, R. A., Naik, V., Zhang, Y., & Adelman, Z. Co-benefits of global greenhouse gas mitigation for future air quality and human health. Nature Climate Change. 2013; 3: 885–9.

The authors should do a comprehensive review of earlier literature and discuss how this work improves earlier studies (which I am not sure if this paper does have any novelty other than repeating the model run). Other comments,

Figure 1: Need a reference for the global emission and climate data sources

Figure 2: It is not possible to read and differentiate all the color lines (there are nearly 40 of them). I suggest only reporting the total of each emission category.

Line 106: It appears that the V4a, V5, V5a are versions of the GAINS Model, not scenarios (see Table 1). Throughout this paper, the authors refer these three abbreviations as "scenarios", which is quite misleading.

Line 109-110: Need a reference for this information.

Line 112: 9 air pollutants: why did not include all the 10 air pollutants discussed earlier in this section?

Line 114: "perfect public measures" - how can policy measures be claimed as "perfect"?

Line 117: "the confusion of air pollution" - what does it mean?

Line 126: Liu et al (2017) - not sure which reference is this since the end of references are ordered by number

Line 462: "healthy harm" - did the authors mean harm to health?

Section 4.2 (including Figure 5): Obviously part of this section should be moved to the Result section. No method was discussed in the Method section how the authors estimated the effects of PM2.5 on life expectancy loss.

Line 478-479: "human life expectancy loss will rise to more than 100 months in the future" - does the 100 months refer to loss per person, per grid cell, or something else?

Line 226: "Data availability" "All the emission data...can be downloaded on..." These read like a research proposal, not a manuscript.

Finally, the entire paper reads like a text translation using some tool without manual proofreading. 

Author Response

Response to Reviewer 1 Comments

Dear Reviewer:

Thank you for your comments concerning our manuscript, entitled “Response of Global Air Pollution Variations to Climate Change and Its Potential Effects on Human Life Expectancy Loss” (ID: sustainability-492493). Those comments are all valuable and very helpful for improving our paper.

The revised manuscript is resubmitted. In addition, the revised manuscript with change tracking is resubmitted to illustrate the details of revision. The responses to your comments are as follows.

Point 1: My main issue with this paper is its novelty. The paper states that "thus far there has been no study investigating how climate change and geographical environment affect and influence the variations of air pollutants around the globe" (Line 80-81), which is not true. For example:

West, J. J., Smith, S. J., Silva, R. A., Naik, V., Zhang, Y., & Adelman, Z. Co-benefits of global greenhouse gas mitigation for future air quality and human health. Nature Climate Change. 2013; 3: 885–9.

The authors should do a comprehensive review of earlier literature and discuss how this work improves earlier studies (which I am not sure if this paper does have any novelty other than repeating the model run).

Response 1: Accepted and revised. The formulation in Introduction has been revised. And more references are added.

Point 2: Figure 1: Need a reference for the global emission and climate data sources

Response 2: Accepted and revised. The data sources have been added.

Point 3: Figure 2: It is not possible to read and differentiate all the color lines (there are nearly 40 of them). I suggest only reporting the total of each emission category.

Response 3: As we would like to illustrate that some certain specific human activities such as residential and other sectors, manufacturing industries and construction, and road transportation, are the major sources of air pollutants, the primary source of each pollutant has been marked in each sub-graph. In addition, two other reviewers also mentioned that there are too many figures and tables in our manuscript, and suggested that we could delete or simplify the Figure 2. Based on the above, we moved Figure 2 to supplementary information as Figure S1.

Point 4: Line 106: It appears that the V4a, V5, V5a are versions of the GAINS Model, not scenarios (see Table 1). Throughout this paper, the authors refer these three abbreviations as "scenarios", which is quite misleading.

Response 4: Accepted and revised. The three versions mentioned in the manuscript were corrected.

Point 5: Line 109-110: Need a reference for this information.

Response 5: Accepted and revised. The references have been cited.

Drew, S., Johan, C. I. K., Elisabetta, V., Rita, v, D., Markus, A., & Zbigniew, K., et al. (2012). Simultaneously Mitigating Near-Term Climate Change and Improving Human Health and Food Security. Science, 335, 183-189.

Liu, F., Klimont, Z., Zhang, Q., Cofala, J., Zhao, L., & Huo, H., et al. (2013). Integrating mitigation of air pollutants and greenhouse gases in Chinese cities: development of GAINS-City model for Beijing. Journal of Cleaner Production, 58, 25-33.

Point 6: Line 112: 9 air pollutants: why did not include all the 10 air pollutants discussed earlier in this section?

Response 6: The 10 air pollutants and greenhouse gases we discussed earlier in this section included CO2, CH4, BC, CO, NH3, NOx, N2O, PM, SO2 and VOC. Among them, CO2, CH4 and N2O are the greenhouse gases. Our research is focus on air pollutants.        Therefore, we only used GAINS model to simulate the 9 major air pollutants including CO, NH3, NOx, OC, PM2.5, PM10, SO2, BC and VOC.

Point 7: Line 114: "perfect public measures" - how can policy measures be claimed as "perfect"?

Response 7: Accepted and revised. It has been revised into "relatively sound public policy measures".

Point 8: Line 117: "the confusion of air pollution" - what does it mean?

Response 8: Accepted and revised. This is a translation error. The correct formulation is "the challenge from air pollution".

Point 9: Line 126: Liu et al (2017) - not sure which reference is this since the end of references are ordered by number

Response 9: Accepted and revised.

Point 10: Line 462: "healthy harm" - did the authors mean harm to health?

Response 10: We do mean harm to health. And it has been changed to "harm to human health" in the manuscript.

Point 11: Section 4.2 (including Figure 5): Obviously part of this section should be moved to the Result section. No method was discussed in the Method section how the authors estimated the effects of PM2.5 on life expectancy loss.

Response 11: Accepted and revised. Section 4.2 (included Figure 5) has been moved to the Result section as Section 3.4. We used GAINS model to calculate the subsequent impacts of PM2.5 on life expectancy loss. The method has been discussed additionally in Section 2.2.

Point 12: Line 478-479: "human life expectancy loss will rise to more than 100 months in the future" - does the 100 months refer to loss per person, per grid cell, or something else?

Response 12: We considered that the expression of this sentence is not appropriate. And the conclusion can be improved. So we revised the conclusion as follows.

This study explored the response of air pollutant diffusion to climate change, and took nine cities with typical terrain to analyze the mechanism by which meteorological characteristics and geographical matrices can drive air pollution at the site-scale. It has been found that natural conditions, such as climate changes, meteorological characteristics, and topography, play important roles in spatial air pollution fluctuations. This study also simulated the potential effects of air pollution in human health by integrating air pollutant emission simulation model (GAINS) and CMIP5.

In summary, short-term climate warming and drying has significant effects and correlation with spatial fluctuations in the global distribution of major air pollutants even though the major sources of air pollutants are mostly emission from human activities. Moreover, air pollutants emissions show a significant response to the climate change in China but little correlation in developed countries. At a local scale, the dilution and diffusion of air pollutants in a specific area is driven by the thermal and dynamic effects of the local circulation model affected by geographical conditions (such as terrain relief and type of landforms, etc.) and comprehensive meteorological conditions.

Based on the research, we supposed that if no further controls and attentions paid to climate change and air pollution, there will be potential threat in human health. Therefore, governments should adopt corresponding policies and measures to slow down the trend of global climate warming and drying, and pay more attentions to air pollution control policy setting.

Point 13: Line 226: "Data availability" "All the emission data...can be downloaded on..." These read like a research proposal, not a manuscript.

Response 13: Accepted and revised. The formulation in Section 2.1 Data has been revised.

Point 14: Finally, the entire paper reads like a text translation using some tool without manual proofreading. 

Response 14: Accepted and revised. Our manuscript is still under polishing. We have invited a translation company named "EDITAGE" to help us improve the English writing. 

Reviewer 2 Report

Thanks for the authors’ efforts in studying the interesting issue of air pollution and human health. However, too many components are put together in the paper (Global Air Pollution, global climate change, terrain effects on air pollution, pollution links to health, life expectancy, global and China). These sub-topics are loosely linked in the paper. Therefore, I suggest major revisions of the paper, and the authors shall focus on one or two specific topics.

Normally, references are not used in abstract of a scientific paper. The “Fig. 1a” in the abstract can be removed.

Figure 1, what is the source of this figure and the data? There must be references.  The both Y axis of Figure 1b need titles. The both fig 1a and 1b can me smaller. You don’t need the table in 1a.

The big numbers should be wrote in scientific notation, e.g. 1153401.40 and 28532332.37. And, these numbers need references.

The introduction is very weak and has ignored many important information. Two of the authors’ statements are not true “few studies have explored the relationship between the change in distribution of all air pollutants and the climate change from a global perspective”, and the statement “there has been no study investigating how climate change and geographical environment affect and influence the variations of air pollutants around the globe.”. There are indeed many studies on this:

·         Daniel J.Jacob, Darrell A.Winner. Effect of climate change on air quality. Atmospheric Environment, Volume 43, Issue 1, January 2009, Pages 51-63

·         Hannah Hoag. Air quality to suffer with global warming.  Study suggests effects of climate change will slow air circulation around the world. 22 June 2014. NATURE.

·         Silva, et al. Future global mortality from changes in air pollution attributable to climate change. Nature Climate Change volume7, pages647–651 (2017)

About the links between air pollution and health, there are also some important studies that should be referred.

·         LE Yang, P Hoffmann, J Scheffran. Health impacts of smog pollution: The human dimensions of exposure. The Lancet Planetary Health 1 (4), e132-e133

·         L Yang, P Hoffmann, J Scheffran, S Rühe, J Fischereit, I Gasser. An Agent-Based Modeling Framework for Simulating Human Exposure to Environmental Stresses in Urban Areas. Urban Science 2 (2), 36

GAINS and CMIP5 are the major models used in this study, but there are not any references about the two models. And there is not any introduction to CMIP5. This is not acceptable in a scientific paper.

Figure 2 is hard to read, see and understand. The many colors of the lines can not be recognized. Actually I didn’t find the value of using this figure. The authors may delete the figure or simplify it significantly.

Figure 3, the legends for all sub-graphs are the same. Thus you can use one for all. In addition, what is GADD? You have never mentioned it in the text.

There are too many figures and tables in the paper. Some of them can be deleted or moved to a supplementary file. For example, the Extended data figure 1-6, table 3, table 4, extended table 1.

The section “4 discussion” has very little links to the formers sections. The discussion is actually an independent analysis of weather conditions and terrains, but not discussion of the former studies.

The figure 5 comes very suddenly and is not well connected to other parts of the paper.  And the methods and data for calculating life expectancy loss are not clear.

The conclusion part is very coarse. The findings should be carefully treated when generalizing them to common senses. For instance, the concluded points in China are not necessarily right for other countries. “… human life expectancy loss will rise to more than 100 months in the future. Therefore, all countries should …”

Overall, the paper is too ambitious to deal with so many components, and each parts of the components are not deeply investigated. It is strongly suggested that the authors should focus on one or two specific topics in one paper.

Author Response

Response to Reviewer 2 Comments

Dear Reviewer:

Thank you for your comments concerning our manuscript, entitled “Response of Global Air Pollution Variations to Climate Change and Its Potential Effects on Human Life Expectancy Loss” (ID: sustainability-492493). Those comments are all valuable and very helpful for improving our paper.

The revised manuscript is resubmitted. In addition, the revised manuscript with change tracking is resubmitted to illustrate the details of revision. The responses to your comments are as follows.

Point 1: Thanks for the authors’ efforts in studying the interesting issue of air pollution and human health. However, too many components are put together in the paper (Global Air Pollution, global climate change, terrain effects on air pollution, pollution links to health, life expectancy, global and China). These sub-topics are loosely linked in the paper. Therefore, I suggest major revisions of the paper, and the authors shall focus on one or two specific topics.

Response 1: Accepted and revised. The structure of our manuscript has been adjusted and revised.

Point 2: Normally, references are not used in abstract of a scientific paper. The “Fig. 1a” in the abstract can be removed.

Response 2: Accepted and Revised. The “Fig. 1a” in the abstract has been removed.

Point 3: Figure 1, what is the source of this figure and the data? There must be references.  The both Y axis of Figure 1b need titles. The both fig 1a and 1b can me smaller. You don’t need the table in 1a.

Response 3: Accepted and revised. The data sources have been added. The titles of both Y axis of Figure 1b have been added. And both fig 1a and 1b have been revised to be smaller.

Point 4: The big numbers should be wrote in scientific notation, e.g. 1153401.40 and 28532332.37. And, these numbers need references.

Response 4: Accepted and Revised.

Point 5: The introduction is very weak and has ignored many important information. Two of the authors’ statements are not true “few studies have explored the relationship between the change in distribution of all air pollutants and the climate change from a global perspective”, and the statement “there has been no study investigating how climate change and geographical environment affect and influence the variations of air pollutants around the globe.”. There are indeed many studies on this:

·         Daniel J.Jacob, Darrell A.Winner. Effect of climate change on air quality. Atmospheric Environment, Volume 43, Issue 1, January 2009, Pages 51-63

·         Hannah Hoag. Air quality to suffer with global warming.  Study suggests effects of climate change will slow air circulation around the world. 22 June 2014. NATURE.

·         Silva, et al. Future global mortality from changes in air pollution attributable to climate change. Nature Climate Change volume7, pages647–651 (2017)

About the links between air pollution and health, there are also some important studies that should be referred.

·         LE Yang, P Hoffmann, J Scheffran. Health impacts of smog pollution: The human dimensions of exposure. The Lancet Planetary Health 1 (4), e132-e133

·         L Yang, P Hoffmann, J Scheffran, S Rühe, J Fischereit, I Gasser. An Agent-Based Modeling Framework for Simulating Human Exposure to Environmental Stresses in Urban Areas. Urban Science 2 (2), 36

Response 5: Accepted and revised. The formulation in Introduction has been revised. And the references the reviewer has mentioned are added.

Point 6: GAINS and CMIP5 are the major models used in this study, but there are not any references about the two models. And there is not any introduction to CMIP5. This is not acceptable in a scientific paper.

Response 6: Accepted and revised.

The references about GAINS model and CMIP5 have been cited. And the website of the GAINS model has been mentions. We used CMIP5 as global climate data in our study. The related introduction has been added in Section 2.1 Data. The additional references and website are as follows.

Drew, S., Johan, C. I. K., Elisabetta, V., Rita, v, D., Markus, A., & Zbigniew, K., et al. (2012). Simultaneously Mitigating Near-Term Climate Change and Improving Human Health and Food Security. Science, 335, 183-189.

Liu, F., Klimont, Z., Zhang, Q., Cofala, J., Zhao, L., & Huo, H., et al. (2013). Integrating mitigation of air pollutants and greenhouse gases in Chinese cities: development of GAINS-City model for Beijing. Journal of Cleaner Production, 58, 25-33.

http://www.iiasa.ac.at/web/home/research/researchPrograms/air/Global_emissions.html

Taylor, F. E., Stouffer, R. J., & Meehl, G. A. (2012). An overview of CMIP5 and the experiment design. Bull American Meteorological Society, (4), 485-498.

Point 7: Figure 2 is hard to read, see and understand. The many colors of the lines can not be recognized. Actually I didn’t find the value of using this figure. The authors may delete the figure or simplify it significantly.

Response 7: Accepted and revised. As we would like to illustrate that some certain specific human activities such as residential and other sectors, manufacturing industries and construction, and road transportation, are the major sources of air pollutants, the primary source of each pollutant has been marked in each sub-graph. Based on the above, we moved Figure 2 to supplementary information as Figure S1.

Point 8: Figure 3, the legends for all sub-graphs are the same. Thus you can use one for all. In addition, what is GADD? You have never mentioned it in the text.

Response 8: The legends for all sub-graphs are not totally the same. When GADD1, the legends include following four elements: 0, 0 – GADD, GADD – 1, 1. When GADD1, the legends include following four elements: 0, 0 – 1, 1 – GADD, GADD.

      GADD is the acronym of Global Average Dynamic Degree. It refers to the average value of all the units’ spatial dynamic degree in each sub-graph. It is now mentioned in Figure 3.

Point 9: There are too many figures and tables in the paper. Some of them can be deleted or moved to a supplementary file. For example, the Extended data figure 1-6, table 3, table 4, extended table 1.

Response 9: Accepted and revised. The Extended data figure 1-6 have been moved to supplementary file as Figure S16-21. Table 3 and table 4 have been moved to supplementary file as Table S1 and Table S2. Extended table 1 has been moved to supplementary file as Table S5.

Point 10: The section “4 discussion” has very little links to the formers sections. The discussion is actually an independent analysis of weather conditions and terrains, but not discussion of the former studies.

Response 10: We analyzed the response of global air pollution to climate change in this manuscript, and further investigated how meteorological characteristics and complex geographical environment variations can drive spatial air pollution variations. We considered that the relationship between air pollutant emissions and geographical conditions is an interesting point. It can help us explore the mechanism of air pollutant diffusion and establish connections between macroeconomic climate change and local scale air circulation.

Point 11: The figure 5 comes very suddenly and is not well connected to other parts of the paper.  And the methods and data for calculating life expectancy loss are not clear.

Response 11: Accepted and revised. We used GAINS model to calculate the subsequent impacts of PM2.5 on life expectancy loss. The method has been discussed additionally in Section 2.2. We aimed to explore the response of global air pollution to climate change and its potential effects on human life expectancy loss. The calculation of impact of PM2.5 to human life expectancy loss is based on the simulating results of emissions of air pollutants by using GAINS model. Section 3.3 is the basis of Section 3.4.

Point 12: The conclusion part is very coarse. The findings should be carefully treated when generalizing them to common senses. For instance, the concluded points in China are not necessarily right for other countries. “… human life expectancy loss will rise to more than 100 months in the future. Therefore, all countries should …”

Response 12: Accepted and revised. The conclusion has been revised as follows.

This study explored the response of air pollutant diffusion to climate change, and took nine cities with typical terrain to analyze the mechanism by which meteorological characteristics and geographical matrices can drive air pollution at the site-scale. It has been found that natural conditions, such as climate changes, meteorological characteristics, and topography, play important roles in spatial air pollution fluctuations. This study also simulated the potential effects of air pollution in human health by integrating air pollutant emission simulation model (GAINS) and CMIP5.

In summary, short-term climate warming and drying has significant effects and correlation with spatial fluctuations in the global distribution of major air pollutants even though the major sources of air pollutants are mostly emission from human activities. Moreover, air pollutants emissions show a significant response to the climate change in China but little correlation in developed countries. At a local scale, the dilution and diffusion of air pollutants in a specific area is driven by the thermal and dynamic effects of the local circulation model affected by geographical conditions (such as terrain relief and type of landforms, etc.) and comprehensive meteorological conditions.

Based on the research, we supposed that if no further controls and attentions paid to climate change and air pollution, there will be potential threat in human health. Therefore, governments should adopt corresponding policies and measures to slow down the trend of global climate warming and drying, and pay more attentions to air pollution control policy setting.

Point 13: Overall, the paper is too ambitious to deal with so many components, and each parts of the components are not deeply investigated. It is strongly suggested that the authors should focus on one or two specific topics in one paper.

Response 13: Accepted and revised.

Reviewer 3 Report

Review of: “Response of Global Air Pollution Variations to Climate Change and Its Potential Effects on Human Life Expectancy Loss” by Cheng et al.

The authors aimed ambitiously to examine the effect of climate change on global air pollution and associated health loss, using integrated datasets and various methods. However, the current form of manuscript reads far away from its expectations, since this work only examined the relationship between pollutant emissions and climate indicators. There is seemingly a misunderstanding by the authors on air pollutant emissions and pollutant concentrations. As such, I suggest the authors go back to only talk about the relationship of pollutant emissions with climate indicators and geographical types, etc.

General comments:

-The reasons for correlation between pollutant emissions and climate are not well defended. Basically, only emissions of NH3 and biogenic VOC are dependent on meteorological variables. It is hard to understand why other pollutants mainly emitted from anthropogenic sources are correlated with meteorology. Some correlation is likely due to the common sources that emit air pollutant emissions and greenhouse gas emissions simultaneously.

-The relationship between emissions and geographical conditions is a good point. I suggest the authors can more discussion on this.

Specific comments:

-Fig.2 is not readable from so much colors. And I think this figure is less informative in the main text.

-GAINS model. Do the authors apply the GAINS model to simulate the concentrations of air pollutants?

-Section 2.6 Data. I think data information should put in the beginning of Section 2.

- What exposure functions used to estimate the effect of PM2.5 on human life expectancy? Here PM2.5 refers to emissions or concentrations?

Author Response

Response to Reviewer 3 Comments

Dear Reviewer:

Thank you for your comments concerning our manuscript, entitled “Response of Global Air Pollution Variations to Climate Change and Its Potential Effects on Human Life Expectancy Loss” (ID: sustainability-492493). Those comments are all valuable and very helpful for improving our paper.

The revised manuscript is resubmitted. In addition, the revised manuscript with change tracking is resubmitted to illustrate the details of revision. The responses to your comments are as follows.

Point 1: The reasons for correlation between pollutant emissions and climate are not well defended. Basically, only emissions of NH3 and biogenic VOC are dependent on meteorological variables. It is hard to understand why other pollutants mainly emitted from anthropogenic sources are correlated with meteorology. Some correlation is likely due to the common sources that emit air pollutant emissions and greenhouse gas emissions simultaneously.

Response 1: Accepted and revised.

Point 2: The relationship between emissions and geographical conditions is a good point. I suggest the authors can more discussion on this.

Response 2: Accepted and improved.

Point 3: Fig.2 is not readable from so much colors. And I think this figure is less informative in the main text.

Response 3: As we would like to illustrate that some certain specific human activities such as residential and other sectors, manufacturing industries and construction, and road transportation, are the major sources of air pollutants, the primary source of each pollutant has been marked in each sub-graph. In addition, two other reviewers also mentioned that there are too many figures and tables in our manuscript, and suggested that we could delete or simplify the Figure 2. Based on the above, we moved Figure 2 to supplementary information as Figure S1.

Point 4: GAINS model. Do the authors apply the GAINS model to simulate the concentrations of air pollutants?

Response 4: We applied the GAINS model to simulate the total emissions of air pollutants.

Point 5: Section 2.6 Data. I think data information should put in the beginning of Section 2.

Response 5: Accepted and Revised. Data information has been put in the beginning of Section 2 as Section 2.1.

Point 6: What exposure functions used to estimate the effect of PM2.5 on human life expectancy? Here PM2.5 refers to emissions or concentrations?

Response 6: Accepted and revised. We used GAINS model to calculate the subsequent impacts of PM2.5 on human life expectancy loss. The method has been discussed additionally in Section 2.2. Here PM2.5 refers to emissions.

Round 2

Reviewer 1 Report

Thanks for the authors responses. The paper has been improved. I still don't get the method the authors used to estimate life expectancy loss in the future, other than the added statement "to calculate the subsequent impacts of PM2.5 on human life expectancy loss in the future", including: the equation used, variables included, what was the exposure-response function(s), how the baseline data were collected. I consider the method is key information of the paper and needs to be clearly presented.

Author Response

Response to Reviewer 1 Comments

Dear Reviewer:

Thank you for your comments concerning our manuscript, entitled “Response of Global Air Pollution Variations to Climate Change and Its Potential Effects on Human Life Expectancy Loss” (ID: sustainability-492493). Those comments are all valuable and very helpful for improving our paper.

The further revised manuscript is resubmitted. In addition, the revised manuscript with change tracking is resubmitted to illustrate the details of revision. The responses to your comments are as follows.

Comments and Suggestions for Authors Thanks for the authors responses. The paper has been improved. I still don't get the method the authors used to estimate life expectancy loss in the future, other than the added statement "to calculate the subsequent impacts of PM2.5 on human life expectancy loss in the future", including: the equation used, variables included, what was the exposure-response function(s), how the baseline data were collected. I consider the method is key information of the paper and needs to be clearly presented.

Response: Thank you for your comments and suggestions. The method you have mentioned has been replenished. The introduction has been further revised. The conclusion has also been improved.

The added method is as follows:

The GAINS model was designed to track the air pollutants based on various of economic activities, such as energy consummation, industrial production and agriculture[42]. These economic activities were the main driving forces of air pollutants formation which were selected as the basis of the emission scenarios setting. Different with other air pollution simulation models, the characters of specific regions and sources of air pollutants were considered in air pollution evaluation. Then, technical and non-technical measures were applied to evaluate the potential and cost of air pollutants emission reduction, simulate the emissions accumulation and dispersion process and calculate the impact indicators of air pollution on human health. Depending on the correlation between the indicators that reflect the impacts of air pollutant on human life expectancy and PM 2.5, the human life expectancy loss in the future can be simulated under different scenarios by using GAINS model.

[42] Klimont, Z., Kupiainen, K., Heyes, C., Purohit, P., Cofala, J., Rafaj, P., Borken-Kleefeld, J. and Schöpp, W. (2017). Global anthropogenic emissions of particulate matter including black carbon. Atmos. Chem. Phys., 17, 8681-8723.

Reviewer 2 Report

Thanks for the authors’ revisions and improvements. My comments have been well treated and no further comments come up. I would suggest the paper be accepted for publishing after some minor modifications such as to polish language, remove color in table 1.

Author Response

Response to Reviewer 2 Comments

Dear Reviewer:

Thank you for your comments concerning our manuscript, entitled “Response of Global Air Pollution Variations to Climate Change and Its Potential Effects on Human Life Expectancy Loss” (ID: sustainability-492493). Those comments are all valuable and very helpful for improving our paper.

The further revised manuscript is resubmitted. In addition, the revised manuscript with change tracking is resubmitted to illustrate the details of revision. The responses to your comments are as follows.

Comments and Suggestions for Authors
Thanks for the authors’ revisions and improvements. My comments have been well treated and no further comments come up. I would suggest the paper be accepted for publishing after some minor modifications such as to polish language, remove color in table 1.

Response: Thank you for your comments and suggestions. We have further polished language, and removed the color in Table 1. The introduction has been further revised. The method has been refined. The conclusion has also been improved.

Reviewer 3 Report

  As I mentioned in my first review, the authors mistake the concept of emissions and air pollutant concentrations. The current manuscript is far away from what it is entitled “Response of Global Air Pollution Variations to Climate Change and Its Potential Effects on Human Life Expectancy Loss”.  The paper failed to demonstrate why emissions can be linked with climate change, and the linkage between emissions and health impacts is also wired. I suggest the authors should go through related literature and summarize what’s new from this study.

Author Response

Response to Reviewer 3 Comments

Dear Reviewer:

Thank you for your comments concerning our manuscript, entitled “Response of Global Air Pollution Variations to Climate Change and Its Potential Effects on Human Life Expectancy Loss” (ID: sustainability-492493). Those comments are all valuable and very helpful for improving our paper.

The further revised manuscript is resubmitted. In addition, the revised manuscript with change tracking is resubmitted to illustrate the details of revision. The responses to your comments are as follows.

Comments and Suggestions for Authors
As I mentioned in my first review, the authors mistake the concept of emissions and air pollutant concentrations. The current manuscript is far away from what it is entitled “Response of Global Air Pollution Variations to Climate Change and Its Potential Effects on Human Life Expectancy Loss”.  The paper failed to demonstrate why emissions can be linked with climate change, and the linkage between emissions and health impacts is also wired. I suggest the authors should go through related literature and summarize what’s new from this study.

Response: Thank you for your comments and suggestions. The introduction has been further revised. We have replenished some important demonstration and references to illustrate the linkage between air pollution emissions and climate change. We have also summarized what’s new from our study. The method has been refined. The conclusion has also been improved.

The demonstration of why emissions can be linked with climate change is as follows:

Among all sources of air pollution emissions, anthropogenic sources, represented by residential and other sectors, manufacturing industries and construction, and road transportation, are main sources of air pollutants. From 1970 to 2010, the above-mentioned human activities produced a total of 2.8532*107 tons of air pollutants, which was 65.88% of the total emission of the 10 major air pollutants. Existing studies indicate that climate change can affect anthropogenic air pollution emissions to a certain extent. Meanwhile, natural sources, such as wildfires, dust, natural biogenic and lightning emissions, are still nonnegligible sources of air pollution. Up to now, several studies demonstrated that climate change has significant influence in both anthropogenic and natural sources of air pollution emissions.

References supporting this section have been added and as is as follows:

Patrick L. Kinney ScD. (2008). Climate Change, Air Quality, and Human Health. American Journal of Preventive Medicine, 35(5), 459-467.

Raquel, A. S., J. Jason, W., Yuqiang, Z., Susan, C. A., Jean-François, L., Drew, T. S., William, J. C., Stig, D., Greg, F., Gerd, F., Larry, W. H., Tatsuya, N., Vaishali, N., Steven, R., Ragnhild, S., Kengo, S., Toshihiko, T., Daniel, B., Philip, Cameron-Smith., Irene, C., Ruth, M. D., Veronika, E., Beatrice, J., I. A. MacKenzie., David, P., Mattia, R., David, S. S., Sarah, S., Sophie, S. & Guang, Z. (2013). Environmental Research Letters, 8, 034005.

Nakicenovic, N., Alcamo, J., Davis, G., Vries, B. d., Fenhann, J., Gaffin, S., Gregory, K., Grübler, A., Jung, T. Y., Kram, T., Rovere, E. L. L., Michaelis, L., Mori, S., Morita, T., Pepper, W., Pitcher, H., Price, L., Riahi, K., Roehrl, A., Rogner, H.-H., Sankovski, A., Schlesinger, M., Shukla, P., Smith, S., Swart, R., Rooijen, S. v., Victor, N., Dadi, Z. (2000). Emissions Scenarios. A Special Report of IPCC Working Group III; Intergovernmental Panel on Climate Change; Cambridge, UK.

Sanderson, M. G., Jones, C. D., Collins, W. J., Johnson, C. E., Derwent, R. G. (2003). Effect of climate change on isoprene emissions and surface ozone levels. Geophysical Research Letters, 30(18), 1936.

Naik, V., Delire, C., Wuebbles, D. J. (2004). Sensitivity of global biogenic isoprenoid emissions to climate variability and atmospheric CO2. Journal of Geophysical Research, 109, D06301.

Lathiere, J., Hauglustaine, D. A., De Noblet-Ducoudre, N., Krinner, G., Folberth, G. A. (2005). Past and future changes in biogenic volatile organic compound emissions simulated with a global dynamic vegetation model. Geophysical Research Letters, 32, L20818.

Arneth, A., Niinemets, U., Pressley, S., Back, J., Hari, P., Karl, T., Noe, S., Prentice, I. C., Serca, D., Hickler, T., Wolf, A., Smith, B. (2007). Process-based estimates of terrestrial ecosystem isoprene emissions: incorporating the effects of a direct CO2-isoprene interaction. Atmospheric Chemistry and Physics, 7, 31-53.

Spracklen, D. V., Mickley, L. J., Logan, J. A., Hudman, R. C., Yevich, R., Flannigan, M. D., Westerling, A. L. (2009). Impacts of climate change from 2000 to 2050 on wildfire activity and carbonaceous aerosol concentrations in the western United States. Journal of Geophysical Research: Atmospheres, 114(D20), D20301.

Guenther, A. B., Jiang, X., Heald, C. L., Sakulyanontvittaya, T., Duhl, T., Emmons, L. K., Wang, X. (2012). The Model of Emissions of Gases and Aerosols from Nature version 2.1 (MEGAN2.1): an extended and updated framework for modeling biogenic emissions. Geoscientific Model Development, 5, 1471-1492.

Slezakova, K., Morais, S., Pereira, M. d. C. (2013). Forest fires in Northern region of Portugal: Impact on PM levels. Atmospheric Research, 127, 148-153.

Wang, Y., Shen, L., Wu, S., Mickley, L., He, J., Hao, J. (2013). Sensitivity of surface ozone over China to 2000−2050 global changes of climate and emissions. Atmospheric Environment, 75, 374-382.

Arlene, M. F., Vaishali N., & Eric, M. L. (2015). Air Quality and Climate Connections. Journal the Air & Waste Management Association, 65(6), 645-685.

Erika, V. S., Paul, S. M., James, D, A., Lori, B., Piers, F., Dvid, F., Axel, L., William, T. M., Pauli, P., Mattie, R., Katerina, S., & Mark, A. S. (2015). Chemistry and the linkages between air quality and climate change. Chemical Reviews, 115, 3856-3897.

Raquel, A. S., J. Jason, W., Jean-Francois, L., Drew, T. S., William, J. C., Greg, F., Gerd, A. F., Larry, W. H., Tatsuya, N., Vaishali, N., Steven, T. R., Kengo, S., Toshihiko, T., Daniel, B., Philip, C. S., Ruth, M. D., Beatrice, J., Ian, A. M., David, S. S., & Guang, Z. (2017). Future global mortality from changes in air pollution attributable to climate change. Nature Climate Change, (7), 647-651.

Daniel, J. J., & Darrell, A. W. (2009). Effect of climate change on air quality. Atmospheric Environment, 43, 51-63.

D. V. Spracklen, L. J. Mickley, J. A. Logan, R. C. Hudman, R. Yevich, M. D. Flannigan, & A. L. Westerling. (2009). Impacts of climate change from 2000 to 2050 on wildfire activity and carbonaceous aerosol concentrations in the western United States. Journal of Geophysical Research, 114.

The literature review and the summary of what’s new from our study is as follows:

The available studies which explored the impacts of climate change to air pollution emissions mostly focused on a certain kind of air pollutant or a certain source of air pollution emission. It is necessary to research on the correlation between total air pollution emissions and climate change. Moreover, the existing studies on investigating how climate change and geographical conditions affect and influence the variations of air pollutants around the globe haven’t included a variety of terrain synthetically in one research. Based on the limitation of previous studies, our research explored the correlation between total air pollution emissions and climate change, and focus on the dynamic mechanism that natural factors, especially for local weather conditions and local terrain conditions, combining effect the spatial diffusion of various of air pollutants.

Round 3

Reviewer 3 Report

I thank the efforts made by the authors to improve the paper. There are still two important issues that will prevent its publication. Firstly, I think there is little evidence that climate change can affect anthropogenic emissions, and none of refs. 8, 9, 22, 23 you cited in the text support this linkage. Secondly, the authors should clearly use the words like “air pollutant emissions” rather than “air pollution” or “air pollutants” in the title and throughout the whole text where applicable. Because you just talk about emissions, any usage of “air pollution” or “air pollutants” here is inappropriate. I will recommend publication of this paper under the condition that at least these two points are addressed. Also please check over you reference list, some references should give the last name rather than first name of the authors (e.g., 9, 19, 20, 21, 22).

Author Response

Response to Reviewer 3 Comments

Dear Reviewer:

Thank you for your comments concerning our manuscript, entitled “Response of Global Air Pollution Variations to Climate Change and Its Potential Effects on Human Life Expectancy Loss” (ID: sustainability-492493). Those comments are all valuable and very helpful for improving our paper.

The further revised manuscript is resubmitted. In addition, the revised manuscript with change tracking is resubmitted to illustrate the details of revision. The responses to your comments are as follows.

Comments and Suggestions for Authors
I thank the efforts made by the authors to improve the paper. There are still two important issues that will prevent its publication. Firstly, I think there is little evidence that climate change can affect anthropogenic emissions, and none of refs. 8, 9, 22, 23 you cited in the text support this linkage. Secondly, the authors should clearly use the words like “air pollutant emissions” rather than “air pollution” or “air pollutants” in the title and throughout the whole text where applicable. Because you just talk about emissions, any usage of “air pollution” or “air pollutants” here is inappropriate. I will recommend publication of this paper under the condition that at least these two points are addressed. Also please check over you reference list, some references should give the last name rather than first name of the authors (e.g., 9, 19, 20, 21, 22).

Response: Thank you very much for your suggestions. We have accepted and revised our manuscript according to your precious comments. We checked all the references we cited, and revised the inappropriate expression. We have also changed the words like “air pollution” and “air pollutants” into “air pollutant emissions” in our paper. In addition, the reference list has been carefully checked and revised.
